# The Role of Canopy Cover Dynamics over a Decade of Changes in the Understory of an Atlantic Beech-Oak Forest

**Mercedes Valerio** [1,*] , **Ricardo Ibáñez** [1] and **Antonio Gazol** [2]

1   Departamento de Biología Ambiental, Facultad de Ciencias, Universidad de Navarra, Calle Irunlarrea 1, 31008 Pamplona, Spain; ribanez@unav.es
2   Instituto Pirenaico de Ecología (CSIC), Avenida Montañana 1005, 50059 Zaragoza, Spain; agazolbu@gmail.com
*   Correspondence: mvalerio.1@alumni.unav.es

**Abstract:** The understory of temperate forests harbour most of the plant species diversity present in these ecosystems. The maintenance of this diversity is strongly dependent on canopy gap formation, a disturbance naturally happening in non-managed forests, which promotes spatiotemporal heterogeneity in understory conditions. This, in turn, favours regeneration dynamics, functioning and structural complexity by allowing changes in light, moisture and nutrient availability. Our aim is to study how gap dynamics influence the stability of understory plant communities over a decade, particularly in their structure and function. The study was carried out in 102 permanent plots (sampled in 2006 and revisited in 2016) distributed throughout a 132 ha basin located in a non-managed temperate beech-oak forest (Bertiz Natural Park, Spain). We related changes in the taxonomical and functional composition and diversity of the understory vegetation to changes in canopy coverage. We found that gap dynamics influenced the species composition and richness of the understory through changes in light availability and leaf litter cover. Species with different strategies related to shade tolerance and dispersion established in the understory following the temporal evolution of gaps. However, changes in understory species composition in response to canopy dynamics occur at a slow speed in old-growth temperate forests, needing more than a decade to really be significant. The presence of gaps persisting more than ten years is essential for maintaining the heterogeneity and stability of understory vegetation in old-growth temperate forests.

**Keywords:** community ecology; understory species; canopy gap; niche; temporal variation; seed bank; gap dynamics

## 1. Introduction

Forests cover more than 30% of the Earth's surface and harbour up to 80% of the global terrestrial living carbon storage [1]. Temperate deciduous forests are one of the most abundant woodlands in different regions of the world, where centuries of land reclamation make well-preserved forests scarce [2,3]. Probably as a consequence of their natural value, these forests have been widely studied in Europe, and there is extensive literature about the effect of disturbances and habitat heterogeneity on forest structure and dynamics [4–12]. However, the number of studies decreases as we move southwards. In Northern Spain, where temperate forests find their southern distribution limit, studies of temperate forests are scarce (e.g., [13]), and even more if we consider factors maintaining understory vegetation, which has been understudied (but see [14,15]). It is expected that small-scale disturbances, such as canopy gaps, create environmental heterogeneity [16] that facilitates the development of a heterogeneous understory [17], creates temporal windows for the establishment of species from the soil seed bank [18] and thus increases species coexistence.

Environmental heterogeneity is a key factor maintaining the plant diversity in the understory of temperate forests [7]. In this regard, small-scale disturbances, such as canopy

gaps caused by the death of one or more trees, disrupt the monotony in the overstory, generating variations in light penetration through the canopy and increasing light availability [10,17,19–21]. Moreover, when a gap is formed, the increase in light and the decrease in litter cover can influence the microbial decomposition of the soil organic layer and thus soil pH [22]. Following this, the changes in light penetration and associated changes modulate other factors such as soil water content [19]. All these changes promoted by canopy gaps might potentially impact the understory vegetation [16], which is composed of species that vary, especially in their light requirements [20,23]. Thus, canopy gaps favour the development of the herbaceous and the shrub layer [14] and increase species richness by enabling the establishment of light-demanding species [20] from nearby sites or of species that survive in the seed bank and respond rapidly to the new gaps (i.e., *Rubus ulmifolius* Schott, *Hypericum androsaemum* L. [18]). These species might have potential differences in leaf traits and height from those species found beneath the canopy [19,24]. In this sense, it can be expected that canopy gaps can also increase functional diversity by enabling species coexistence in the understory [25].

In addition, gaps are important for facilitating the regeneration of canopy trees [17]. According to [26], the gap phase is fundamental for canopy regeneration because the mature canopy experiences so much disturbance that the site is opened up, permitting the release of either advance regeneration [27] or the recruitment of new regeneration [28]. In this regard, certain gap characteristics such as gap size, shape and even the plot position within a gap have also been found to be important factors in tree regeneration, having different effects depending on species and life stages [27,28]. In the temperate forests of Northern Spain, periodical small-gap creation by natural disturbances has been found to facilitate the regeneration and growth of beech individuals (*Fagus sylvatica* L.) [29]. Similar results have been found in beech forests in Slovakia [17]. This is because suppressed seedlings in the understory can rapidly become new canopy trees, playing a key role in the dynamics of these forests. In temperate mixed beech–oak forests, beech seedlings can survive in the understory for many years, while oak seedlings can only persist for a short period of time [30]. In Northern Spain, pedunculate oak (*Quercus robur* L.) can be considered a gap-obligate species for its regeneration (e.g., [29]), while in other regions, seedlings can establish satisfactorily in pure oak stands [31].

However, it is important to note that not only the presence or absence of gaps but also their temporal evolution over time, related to their size and age, can have important implications for the forest understory [16]. Small gaps can be rapidly closed by the lateral spread of nearby dominant trees, especially beech, as it is a very plastic species, able to respond quickly to canopy changes [32,33]. In contrast, large gaps will require the establishment of saplings from the understory and will remain open for more time [17,34]. During the time elapsing from gap creation to canopy closure, the forest understory undergoes changes in species composition and richness, which can even lead to plant functional changes [12,16,35,36]. However, little is known on the persistence of such changes over time [12].

In this study, we aim to fill a research gap in southern European temperate deciduous forests by studying the response of vegetation to the opening, closure or persistence of canopy gaps in a temporal window of a decade. We sampled a mixed beech–oak temperate forest in 2006, and we revisited the forest in 2016. In the first census, we established 102 20 × 20 m$^2$ permanent plots in which vegetation was described, and canopy cover and the main environmental conditions were measured. When the plots were revised, we re-surveyed the vegetation and measured the canopy cover again. Thus, we aim to study how the changes in canopy openings and closures have affected the taxonomical and functional characteristics of the understory. We want to: (i) describe gap dynamics in a representative mixed beech-oak temperate forest and (ii) evaluate the effect of gap dynamics on the taxonomical and functional composition and diversity of the understory. We expect that gap dynamics will promote changes in understory species composition, together with an increase in species richness and diversity as a result of gap opening. In

addition, we expect gaps to promote the establishment of species with certain functional traits, leading to changes in understory functional composition and diversity.

## 2. Materials and Methods

### 2.1. Study Site

The study area (cuenca del Suspiro, Señorío de Bertiz Natural Park) is located in the north of Navarra, Spain (43°09′N, 1°37′W). It is a bowl-shaped forest of 132 ha open to the east. The elevation ranges from 200 to 600 m. Geologically, the bedrock belongs to the Palaeozoic era and is mainly composed of silicic shales and schists. The climate is humid-oceanic. Mean annual precipitation is 1564 mm, and mean monthly temperature ranges from 6.8 °C in January to 21.1 °C in August, with an annual mean of 13.7 °C (Spanish Meteorological Agency; Bertiz station; records 1992–2020).

The vegetation is dominated by an acidophilous *Fagus sylvatica* L. forest (European beech) on the north-facing slopes and by a mixed forest with *Quercus robur* L. (pedunculate oak) and *Quercus petraea* (Matt.) Liebl. (sessile oak) on the south-facing ones. Gaps and streams are common throughout the forest. Common species in the understory are the shrubs *Erica vagans* L., *Hypericum androsaemum* and *Ruscus aculeatus* L.; the ferns *Blechnum spicant* (L.) Roth and *Pteridium aquilinum* (L.) Kuhn; and small herbs such as *Deschampsia flexuosa* (L.) Trin. and *Veronica chamaedrys* L. The study site has three important characteristics: it is a continuous forest included in a larger forested landscape, it is structurally heterogeneous (supporting environmental heterogeneity) and it has not been managed for at least 60 years. Nomenclature follows [37] except for *Asteraceae* and *Gramineae*, which follows [38].

### 2.2. Study Design and Variable Survey

The collection of data was based on a stratified sampling approach, using a regular grid of 120 × 120 m$^2$ as strata. The grid contained 102 cells. Inside each cell of the grid, a square sample plot with sides of 20 m was randomly selected (Figure 1). Plots were permanently marked. From May to July 2006, we identified all the vascular plant species present in the understory (≤5 m height) of each plot and assessed their abundance using the Braun-Blanquet cover-abundance scale [39]. In June and July 2016, the 102 original plots were revisited. We re-surveyed the understory vegetation identifying all vascular plant species present in each plot and assessing their abundance as percentage cover. In order to ensure data compatibility, abundance data from 2006 were transformed from Braun-Blanquet scale to percentage scale following [40], and the percentage scale was adjusted between both years. We verified that both abundances were comparable by testing for relationships between the abundance of the most frequent species in 2006 and in 2016, finding a significant and positive relationship for all the species tested (Supporting Information, Table S1). In total, 99 different species were recorded over both years, 92 species in 2006 and 82 species in 2016 (Table S2). The abundance data of understory species in 2006 and 2016 were used to calculate the taxonomical diversity of each plot and year in the form of species richness and Shannon-Wiener index of diversity, which is calculated as $H' = -\sum_{i=1}^{S}(pi \ln pi)$; and combines species richness ($S$) and the proportional abundance of each species ($pi$).

Canopy cover was assessed both in 2006 and 2016, estimating the proportion of the plot covered by the canopy trees (in percentage). We defined canopy gaps as openings in the canopy layer with a cover lower or equal to 80% that were created by the death of one or more trees [41,42]. We considered that a plot had a gap when the vertical projection of the gap fell on the plot [33,42]. The identification of canopy gaps was carried out directly in the field (Figure S1) and was based on expert criteria (R. Ibáñez and A. Gazol, personal observations). To confirm that the canopy openings really were created by tree death and not by the presence of streams or other topographic factors, we checked for the presence of dead tree remains (i.e., fallen logs or stumps) [41] and used aerial photographs to check for tree fall events (Figure S2). The reason for choosing an 80% threshold to define canopy

gaps was based on the fact that all the gaps identified as such in the field had an opening in the canopy greater than 20%. In contrast, for canopy openings smaller than 20%, we could not guarantee that the cause of lower canopy coverage was actually tree death. In addition, we were more interested in the effects of intensive disturbance (i.e., large gaps), which seems to have a greater influence on forest dynamics than small gaps, as small gaps tend to close rapidly following beech lateral crown growth [33,43]. Hereafter, we classified plots in four categories according to the presence or absence of gaps in the two years studied (Figure 1), as a proxy of gap dynamics in a decade: plots opened both in 2006 and 2016 (i.e., gap persistence), plots closed both in 2006 and 2016 (i.e., no gaps), plots closed in 2006 but opened in 2016 (i.e., gap opening), and plots opened in 2006 but closed in 2016 (i.e., gap closure).

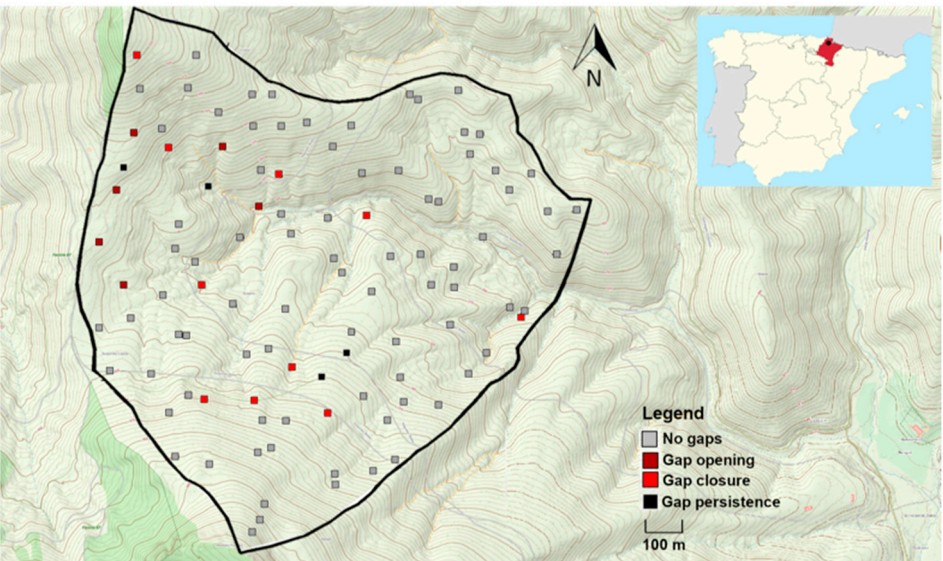

**Figure 1.** Map of the study site showing its location within the Iberian Peninsula and the distribution of the plots throughout the basin. Plots are classified into four categories: no gaps (plots closed both in 2006 and 2016), gap opening (plots closed in 2006 but opened in 2016), gap closure (plots opened in 2006 but closed in 2016) and gap persistence (plots opened both in 2006 and 2016).

We also measured different "environmental" variables (Table S3). The coordinates of the permanent plots were obtained with a GPS device (Garmin Oregon 450), and altitude (in m a.s.l.) was obtained from a topographic map 1:5000 [44]. Slope (sexagesimal degrees) was measured with a clinometer (Silva Clino Master, Silva Sweden). Soil temperature was measured in June 2007 with a soil penetrating thermometer up to a depth of 10 cm (Testo 110, Testo Ltd. Portugal) in each plot. In addition, the temperature was measured every 2 h in three preselected plots (one on the north-facing slope, another on the south-facing one and the last one in the central part of the basin) in order to standardise all the measurements of each plot to avoid differences caused by temperature variation throughout the day. Topsoil moisture (%vol) was measured in July 2007 with a sensor SM200 (Delta-T device, Cambridge UK) at a depth of 51 mm. To obtain both an averaged measure and the CV for soil moisture, various measurements were made in each plot. In particular, both for soil temperature and moisture, 20 measurements were taken throughout each plot following a regular distribution pattern, thus covering the entire plot. Leaf litter cover on the ground (in percentage) was visually estimated in 2006. Radiation and its CV were obtained in 2012 by using hemispheric photography. Photographs were taken at 1 m height, in four points distributed regularly within each plot, and using a digital camera (EOS 50D, Canon, Tokyo, Japan) equipped with a fisheye lens (4.5 mm F2.8 EX DC, Sigma, Tokyo, Japan). Measurements taken on different days were standardised by comparing data with a reference plot on different days and establishing threshold differences. Images were

analysed with Hemiview software (v. 2.1 SR4; 2009 Delta-T Devices, Ltd., Cambridge, United Kingdom), obtaining indirect solar factor (ISF) values, which is an estimation of the indirect radiation levels measured at a site against those measured at an equivalent open-air site [45]. Data for average diameter at breast height (DBH; measured in cm), basal area (in $m^2$ $ha^{-1}$) and Quercus basal area (in percentage) were obtained between July 2016 and February 2018. In each plot, we measured the DBH of the central tree and of the six closest neighbours with a diameter greater than 5 cm, which involves the trees present in more than 80% of the plot area and calculated the mean DBH for each plot. We also measured the distance from the central tree to each of the six neighbours. Then, we calculated the basal plot area (in $m^2$ $ha^{-1}$), dividing the area of the seven trees measured by the area they occupied. *Quercus* basal area is the percentage of the basal area corresponding to oaks (*Quercus robur*, *Quercus petraea* and *Quercus pyrenaica* Willd.).

From April to June 2018, four plant functional traits of ecological importance (plant height, leaf size, specific leaf area and leaf dry matter content; Table S2) were measured for each of the most abundant species in the understory, following the protocols provided by [46,47]. Plant height, the size from ground to the highest photosynthetically active tissue, is often associated with competitive vigour and plant tolerance or avoidance of environmental stress and disturbance [46,47]. Leaf size (LS), the surface area of a whole leaf, tends to be related to competition for light [48], leaf energy and water balance. Moreover, it is negatively associated with drought stress and high-radiation stress [46]. Specific leaf area (SLA), the area of a fresh leaf divided by its oven-dry mass, and leaf dry matter content (LDMC), the oven-dry mass of a leaf divided by its water-saturated fresh mass [47], are negatively and positively correlated, respectively, with resource stress, leaf lifespan and mass-based maximum photosynthetic rate [46]. Plant height was measured in 25 individuals per species, while leaf traits were measured in a total of 20 leaves (two leaves on 10 individuals) per species. Leaves were collected, put in sealed plastic bags with moist paper and stored in a fridge until their processing. Leaf area was measured after scanning the leaves, using the image analysis software ImageJ (v 1.51; [49]). We also weighed the leaves before and after being dried at 80 °C for 48 h to calculate SLA and LDMC. In total, the four functional traits were measured for the 60 most abundant species in the understory. In addition, we obtained data for seed mass (SM) from the TRY database ([50]; Table S4) for 75 of the species studied (Table S2). SM is measured as the oven-dry mass of a seed of a species, and low values of SM are associated with a higher dispersion distance, seed number and higher longevity in seedbanks [46,47].

### 2.3. Data Analysis

Functional composition and diversity indices for each of the five traits studied (plant height, LS, SLA, LDMC, SM) were calculated using the abundance data and trait data matrices. Previous to the calculation of the indices, LS, plant height and SM values were log-transformed to remove skewness. The functional composition was calculated as the community weighted mean (CWM; [51]) of each of the five traits in each plot. The CWM is calculated as $CWM = \sum_{i=1}^{N} p_i \times trait_i$ where $p_i$ is the relative abundance of species $i$, $trait_i$ is the mean trait value of species $i$ and $N$ is the total number of species in the plot or community [51]. For functional diversity, we used two types of indices: functional richness (FRic), measured as the convex hull volume (with the Quickhull algorithm) [52], and the Rao quadratic index [53], measured as the sum of pairwise functional distances between species weighted by relative abundance, and calculated as $Q = \sum_{i-1}^{S-1} \sum_{j=i+1}^{S-1} d_{ij} p_i p_j$ where $d_{ij}$ is the dissimilarity between species $i$ and $j$, $p_i$ and $p_j$ are the relative abundances of species $i$ and $j$, respectively, and $S$ is the total number of species in the plot or community [54]. As both FRic and Rao indices might be affected by species richness, leading to the appearance of significant relationships simply due to random variation in species richness [55], we created null models by randomising (with 999 permutations) species occurrences (for calculating null FRic) and by rearranging the abundances of the species present in the species pool within each plot (for calculating null Rao). For calculating null FRic, we used

presence/absence data. Then, differences between the observed index and null index were calculated by using the standardised effect size (SES; observed-expected/sd(expected)), which measures the number of standard deviations that an observed index is above or below an expected index [56]. As a result, we obtained SESFRic and SESRao indices for which positive and negative values indicate, respectively, higher or lower functional richness and diversity than expected by chance [57].

We looked for patterns and changes in understory species composition from 2006 to 2016 using a non-metric multidimensional scaling ordination (NMDS; [58]) with two dimensions and using Bray-Curtis dissimilarity. We carried out a PERMANOVA analysis [59] to test if differences in species composition between plots were explained by time and by the presence and absence of canopy gaps or their interaction. For a better interpretation of the results, we tested for significant Pearson's correlations between the two axes scores and the environmental variables studied, adding the abundance of *Quercus* and *Fagus* saplings to see if the presence of *Quercus* or *Fagus* in the understory was related to gaps. In addition, and to check which environmental variables were actually associated with canopy gaps, we selected plots with gaps in 2006 and comparable plots without gaps that were located at the same orientation and altitude (a total of 12 north-oriented plots) and used a *t*-test to test for significant differences in the environmental variables studied between plots with and without gaps.

In order to study the effect of gap dynamics over the forest understory vegetation, we tested for significant relationships between the four plot categories previously defined (no gaps, gap closure, gap opening and gap persistence) and changes in the taxonomical and functional composition and diversity of the understory from 2006 to 2016, using the Dunnett's [60] modified Tukey-Kramer (DTK) test. This test is a pairwise multiple comparison procedure that allows testing for mean differences among groups with unequal sample size and variance [61]. In addition to testing for relationships between the four plot categories and understory changes, we also tested for relationships between the four plot categories and the values of understory taxonomical and functional composition and diversity at the end of the decade (in 2016).

Finally, we also studied the effect of gaps at the species level. Firstly, we carried out an indicator species analysis (ISA) using the IndVal index [62] to identify the species that were significantly associated with any of the four categories of plots related to gap dynamics (no gaps, gap opening, gap closure and gap persistence). Then, to see if the species selected as gap indicators shared similar values of functional traits (plant height, LS, SLA, LDMC or SM), we compared their values with those observed for the rest of the species. Thus, for each one of the species selected as indicators of one of the four plot categories, we calculated a standardised effect size as follows: (trait value of indicator species–mean trait values of non-indicator species)/SD of trait values of non-indicator species. We could see the magnitude of the difference in functional trait value between the indicator species and the rest of the species. We considered that the SES showed significant differences between the studied species and the rest of the species when values were higher than $\pm 1.5$ standard deviations.

All analyses were carried out with R statistical environment (v. 4.0.3; [63]). To calculate functional indices and randomisations we used the *dbFD* function ("FD" package; [64,65]) and the *randomizeMatrix* function ("Picante" package; [66]), respectively. NMDS ordination, PERMANOVA and Dunnett modified Tukey–Kramer tests were carried out using the *metaMDS* function, *adonis* function ("vegan" package; [67]) and the *DTK. test* function ("DTK" package; [61]), respectively. Indicator Species Analysis was carried out with the *multipatt* function ("indicspecies" package; [68]).

## 3. Results

### 3.1. Frequency and Dynamism of Gaps

Significant changes in canopy cover occurred in the decade from 2006 to 2016 (Wilcoxon signed-rank test for paired samples: V = 1120, $p$ = 4.292 × $10^{-5}$), suggesting a high dynamism in canopy opening and closure in the temperate forest studied (Figure 2). In particular, we found an increase in tree layer cover from 2006 to 2016 in most plots, with a consequently high rate of canopy closure. In 2006, 14 plots contained canopy gaps. In 2016, ten out of these 14 plots were already closed (gap closure) due to an increase in canopy cover up to 60%. In only four plots, the gaps present in 2006 remained open until 2016, with a canopy cover lower than 80% (gap persistence), although their canopy cover had also increased. In 2016, new gaps appeared in six plots that we classified as closed canopy plots in 2006 (gap opening). In the other 82 plots, there were no gaps in 2006 nor in 2016 (no gaps).

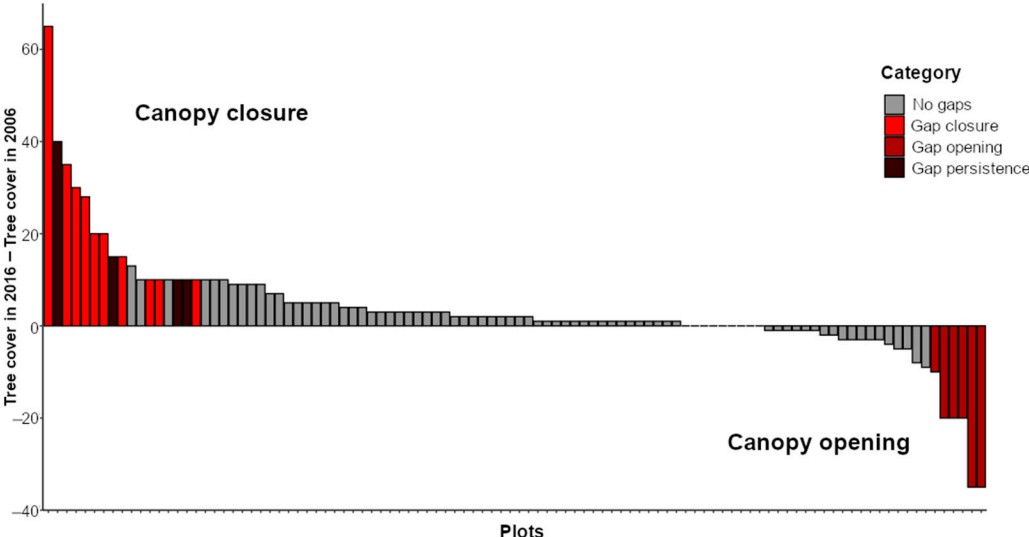

**Figure 2.** Difference in the percentage of tree layer cover in each plot between 2016 and 2006. Plots are ordered from positive differences in tree cover (i.e., canopy closure) to negative differences (i.e., canopy opening), and are classified in four categories: "no gaps" if there are no gaps in 2006 nor in 2016, "gap closure" if a gap present in 2006 is closed in 2016, "gap opening" if a new gap appears from 2006 to 2016 and "gap persistence" if the gap is present both in 2006 and 2016.

### 3.2. Effects of Gap Dynamics on the Understory Taxonomical and Functional Composition and Diversity

The PERMANOVA analysis showed that the differences found in species composition between plots were significantly related to year (F = 4.7353, $p$ = 0.001) and to the presence and absence of gaps (F = 3.8080, $p$ = 0.001), but not to the interaction between both (F = 1.2159, $p$ = 0.267). Plots with persistent gaps, plots with gap opening and even plots that had undergone gap closure seemed to have a different species composition than plots without gaps, tending to be concentrated in negative values of NMDS1 (Figure 3). In this regard, negative values of NMDS1 were significantly related to gaps and to a lower tree layer cover (Figure S3; Table S5). South-facing plots located at low altitudes, with steep slopes, higher light availability and heterogeneity, higher soil temperature, higher abundance of *Quercus* in the canopy and the understory and lower leaf litter cover and basal tree area also had negative scores in NMDS1 (Figure S3; Table S5). In contrast, NMDS2 represented changes in species composition that were not related to a gradient of gap opening or closure but to soil moisture and time (Figure S3; Table S5). In this sense, there were more significant changes over time in the species composition towards NMDS2 (Wilcoxon signed-rank test for paired samples: V = 1319; $p$ = 1.283 × $10^{-5}$) than towards NMDS1 (V = 1861; $p$ = 0.01066). Changes in species composition from 2006 to 2016 in

NMDS2 were directed towards a species composition typical of north-facing communities located at higher altitudes, with steep slopes, high-diameter trees, higher soil moisture and leaf litter cover, lower light availability, soil temperature and a lower abundance of *Quercus* in the canopy and the understory (Figure S3; Table S5).

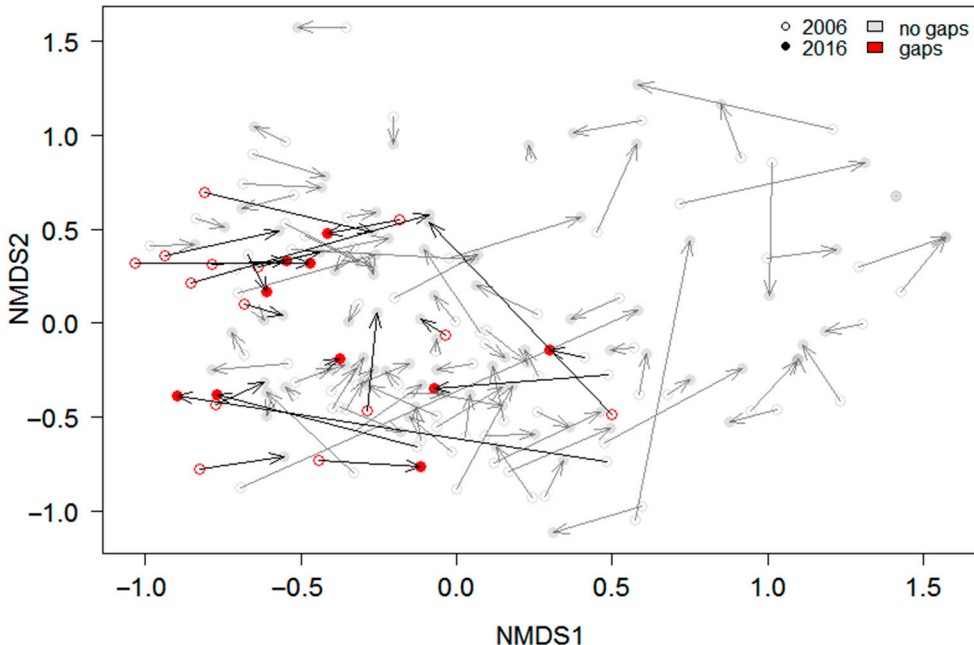

**Figure 3.** NMDS plot showing changes in understory species composition from 2006 to 2016. Circles represent relevé plots. Each arrow connects the same plot in 2006 and 2016. Red colour indicates if the plot has a gap in 2006 (empty circles) or 2016 (filled circles). Grey colour indicates no gap. Stress = 0.2. Iterations = 91.

When looking at the relationship between each plot category (gap closure, opening, persistence or no gaps) and taxonomical and functional changes in the understory from 2006 to 2016, we found that plots with gap persistence had significantly fewer changes in understory species composition towards the NMDS2 than plots without gaps (Table 1). The rest of the studied variables did not differ between categories (Table 1). When instead of focusing on changes, we focused on the values at the end of the decade of the variables studied; we found that plots with gap closure and gap persistence had a significantly different species composition in NMDS1 (towards negative values) than plots without gaps (Figure S4). Although plots with gap opening also tended to have negative values of NMDS1 at the end of the decade, they were not significantly different from any of the other categories (Figure S4). In addition, species richness was significantly higher at the end of the decade in plots that underwent gap opening than in plots without gaps, but no significant differences were found for other pairwise comparisons (Figure S4). The values found at the end of the decade for species composition towards NMDS2, species diversity and functional composition, richness and diversity did not differ between plot categories.

**Table 1.** Results of the DTK test showing the differences between the four categories of plots studied in Bertiz (with gap closure, opening, gap persistence or no gaps) in terms of understory taxonomical and functional changes from 2006 to 2016. The taxonomical variables studied were changes between 2006 and 2016 (dif.) in: the scores of the first (NMDS1) and second (NMDS2) axis of the NMDS ordination, species abundance (abundance), species richness (richness) and Shannon index of species diversity (shannon). The functional variables studied were changes between 2006 and 2016 (dif.) in: the functional composition (measured as the community weighted mean: CWM), functional richness (FRic) and functional diversity (Rao) of leaf dry matter content (LDMC), specific leaf area (SLA), leaf size in its logarithmic form (LSlog), plant height in its logarithmic form (PHlog) and seed mass in its logarithmic form (SMlog). Both functional richness and functional diversity were standardised using the standardised effect size (SES) to avoid spurious results due to changes in species richness. Significant differences are marked in bold ($p < 0.05$).

| Variable | Gap Closure vs. No Gap | Gap Opening vs. No Gap | Gap Persistence vs. No Gap | Gap Closure vs. Gap Opening | Gap Persistence vs. Gap Closure | Gap Persistence vs. Gap Opening |
|---|---|---|---|---|---|---|
| dif.NMDS1 | 0.108 | −0.503 | 0.153 | 0.611 | 0.045 | 0.657 |
| dif.NMDS2 | 0.104 | −0.036 | **−0.129** | 0.140 | −0.233 | −0.093 |
| dif.abundance | 0.863 | 5.380 | 5.213 | −4.517 | 4.35 | −0.167 |
| dif.richness | −2.010 | 5.224 | −1.610 | −7.233 | 0.4 | −6.833 |
| dif.shannon | −0.014 | 0.511 | −0.158 | −0.525 | −0.144 | −0.669 |
| dif.CWMLDMC | 5.757 | 12.018 | 28.618 | −6.261 | 22.861 | 16.600 |
| dif.CWMSLA | 0.667 | 1.461 | 0.899 | −0.794 | 0.232 | −0.562 |
| dif.CWMLSlog | 0.069 | 0.039 | 0.105 | 0.030 | 0.036 | 0.066 |
| dif.CWMPHlog | 0.082 | 0.091 | 0.119 | −0.009 | 0.036 | 0.028 |
| dif.CWMSMlog | 0.292 | 0.047 | 0.412 | 0.245 | 0.120 | 0.365 |
| dif.SESFRicLDMC | −0.025 | −0.219 | −0.489 | 0.193 | −0.464 | −0.270 |
| dif.SESFRicSLA | −0.175 | −0.381 | −0.179 | 0.205 | −0.004 | 0.202 |
| dif.SESFRicLSlog | −0.058 | −0.134 | −0.073 | 0.076 | −0.015 | 0.060 |
| dif.SESFRicPHlog | −0.403 | −0.540 | 1.002 | 0.138 | 1.404 | 1.542 |
| dif.SESFRicSMlog | −1.058 | −0.287 | −0.132 | −0.771 | 0.926 | 0.156 |
| dif.SESRaoLDMC | −0.003 | 0.158 | 0.837 | −0.162 | 0.840 | 0.679 |
| dif.SESRaoSLA | −0.417 | 0.063 | −0.444 | −0.480 | −0.027 | −0.507 |
| dif.SESRaoLSlog | −0.463 | −0.134 | −0.649 | −0.330 | −0.186 | −0.516 |
| dif.SESRaoPHlog | 0.467 | 0.577 | −0.400 | −0.110 | −0.867 | −0.977 |
| dif.SESRaoSMlog | 0.376 | 0.152 | −0.068 | 0.225 | −0.444 | −0.220 |

### 3.3. Indicator Species for Gap Dynamics Categories and Their Functional Characteristics

At the species level, we found that certain species were selected as indicators of a particular category of plot according to gap dynamics, with *Athyrium filix-femina* (L.) Roth and *Campanula patula* L. being related to gap closure *Eupatorium cannabinum* L., *Taraxacum* gr. *officinale* Weber and *Luzula multiflora* (Retz.) Lej. subsp. *multiflora* to gap opening and species such as *Pilosella* sp. and *Veronica officinalis* L. to gap persistence (Table 2). No species were selected as indicators of no gaps.

**Table 2.** List of indicator species for the four categories of plots studied according to gap dynamics (gap closure, gap opening, gap persistence and no gaps). The category "no gaps" does not appear in the table because there were no species significantly selected as indicators of that category. The statistic used was the IndVal index, as this indicates the strength of the association between a species and a certain plot category. Alpha = 0.05 and 999 permutations.

| Canopy Dynamics | Species | Statistic | *p* Value |
|---|---|---|---|
| Gap closure | *Athyrium filix-femina* | 0.628 | 0.041 * |
|  | *Campanula patula* | 0.517 | 0.04 * |
| Gap opening | *Eupatorium cannabinum* | 0.577 | 0.006 ** |
|  | *Taraxacum* gr. *officinale* | 0.567 | 0.018 * |
|  | *Luzula multiflora* subsp. *multiflora* | 0.547 | 0.046 * |
| Gap persistence | *Juncus* gr. *effusus* | 0.684 | 0.007 ** |
|  | *Dryopteris dilatata* | 0.683 | 0.007 ** |
|  | *Pilosella* sp. | 0.500 | 0.039 * |
|  | *Urtica dioica* | 0.500 | 0.045 * |
|  | *Veronica officinalis* | 0.500 | 0.041 * |

Signif. codes: 0 '**' 0.01 '*' 0.05 '.' 0.1 ' ' 1.

Regarding the functional traits of these species (Figure 4, Table 3), we found that in the case of the species selected as indicators of gap closure, *Athyrium filix-femina* tended to have lower values of LDMC, higher values of SLA and LS and higher values of plant height compared to the rest of the species in the understory. On the other hand, *Campanula patula* tended to have lower values of SM. However, the standardised effect sizes of these trait values were not significant. In the case of species related to gap opening, both *Taraxacum* gr. *officinale* and *Luzula multiflora* subsp. *multiflora* had significantly lower values of plant height than non-indicator species. However, they showed opposite patterns for LDMC, SLA and LS, with *Taraxacum* gr. *officinale* having higher values for these three traits, while *Luzula multiflora* subsp. *multiflora* presented lower values compared to the rest of the species. In terms of SM, both *Eupatorium cannabinum* and *Luzula multiflora* subsp. *multiflora* tended to have lower values of SM. Finally, focusing on species associated with gap persistence, we saw that both *Juncus* gr. *effusus* and *Veronica officinalis* tended to have lower values of SLA and LS than non-indicator species, with this difference being significant in the case of *Juncus* gr. *effusus*. In contrast, we found opposite patterns for LDMC and plant height. *Juncus* gr. *effusus* tended to have lower values of LDMC and had significantly higher values of plant height, while *Veronica officinalis* showed a tendency towards higher values of LDMC and had significantly lower values of plant height. Regarding SM, both *Urtica dioica* L. and *Veronica officinalis* tended to have lower SM than non-indicator species of the understory.

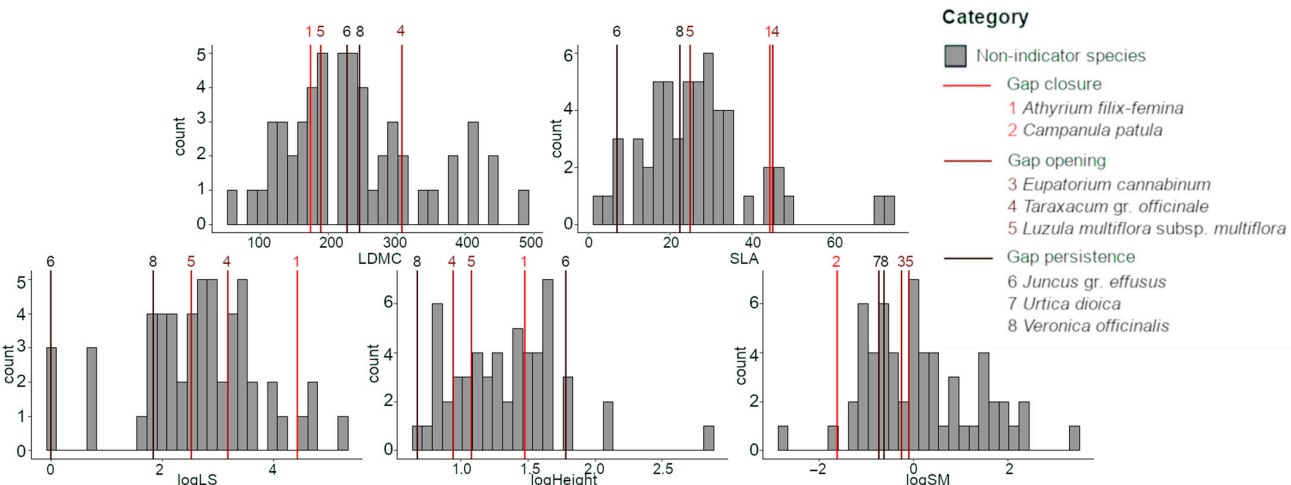

**Figure 4.** Histograms showing the trait values (LDMC: leaf dry matter content, SLA: specific leaf area, logLS: logarithm of leaf size, logHeight: logarithm of plant height, logSM: logarithm of seed mass) of the species found in the understory of the Bertiz forest. Grey bars show the trait values of the species not selected as indicators, while lines indicate the trait values of the species selected as indicators of gap closure, gap opening and gap persistence.

**Table 3.** Standardised effect size of the functional trait values (LDMC: leaf dry matter content, SLA: specific leaf area, logLS: logarithm of leaf size, logHeight: logarithm of plant height, logSM: logarithm of seed mass) of the species selected as indicators of gap closure, gap opening and gap persistence, compared to the trait values of the rest of the species not selected as indicators. Significant values of SES (threshold value = 1.5 × SD) are presented in bold.

| Canopy Dynamics | Indicator Species | LDMC | SLA | logLS | logHeight | logSM |
|---|---|---|---|---|---|---|
| **Gap closure** | *Athyrium filix−femina* | −0.626 | 1.326 | 1.619 | 0.332 | |
| | *Campanula patula* | | | | | −1.514 |
| **Gap opening** | *Eupatorium cannabinum* | | | | | −0.377 |
| | *Taraxacum* gr. *officinale* | 0.676 | 1.376 | 0.469 | **−1.004** | |
| | *Luzula multiflora* subsp. *multiflora* | −0.485 | −0.097 | −0.132 | **−0.664** | −0.257 |
| **Gap persistence** | *Juncus* gr. *effusus* | −0.107 | −1.392 | **−2.445** | **1.087** | |
| | *Urtica dioica* | | | | | −0.779 |
| | *Veronica officinalis* | 0.069 | −0.269 | −0.756 | **−1.671** | −0.691 |

## 4. Discussion

The creation of canopy gaps by the fall of one or more dominant trees as a consequence of natural disturbances induced changes in the understory vegetation of the Bertiz forest. In this mixed beech–oak forest, strict conservation rules have been followed since the middle of the past century allowing the natural dynamics of the forest. The presence of small canopy gaps in 2006, their eventual closure and the appearance of new gaps in 2016 indicates that they represent a natural source of heterogeneity in this forest (Figure 2), as has been found in other forests worldwide [3,34]. In this sense, we found that canopy gaps lead to changes in the environmental conditions of the understory, mainly in terms of light availability and leaf litter cover (Table S6) [19,69]. Through their influence on these environmental conditions, canopy gaps promote a particular plant species composition in the understory of canopy gaps that differs significantly from that found in zones with a closed canopy.

However, what especially influences and modulates the species composition and diversity of the forest understory is not the presence of gaps but rather their temporal evolution. Regarding gap dynamics, we found that in the first years after gap creation, new species establish in the gap, according to the higher levels of species richness found at the end of the decade in plots with gap opening compared to plots without gaps. In this sense, the lack of significant differences between categories in terms of temporal changes in richness from 2006 to 2016, could be due to the high variability of richness values in several closed-canopy plots promoted by the environmental heterogeneity of the forest [15,70]. The reason for the higher species richness in plots with gap openings at the end of the decade can be the greater values of light and lower values of leaf litter found in these plots than in nearby plots with a closed canopy [12,24]. Our results indicate that the occurrence of canopy gaps creates temporal windows that promote the establishment of particular species in the understory. More specifically, plots with recently opened gaps were associated with species such as *Eupatorium cannabinum*, *Taraxacum* gr. *officinale* and *Luzula multiflora* subsp. *multiflora*. These are species related to well-lit sites [71], and their lower values of plant height are consistent with certain studies showing that in the absence of light competition and of a dense herbaceous cover, which are the conditions found in recently opened gaps, shorter species are favoured [72]. In addition, *Eupatorium cannabinum* and *Taraxacum* gr. *officinale* have diaspores (achenes) with a pappus with long hairs that can be easily dispersed by wind [38], which might make these species good early-colonisers of gaps. Species dispersal is an important factor, explaining the distribution of the vegetation in the understory of the studied forests [15], and thus the arrival of diaspores of these light-demanding species from nearby sites can increase the number of species and species coexistence at least in the first years after gap creation. Moreover, the lower seed mass found in all the species selected as indicators with respect to the rest of the species in the understory, although not being significant, is consistent with previous studies showing that a low seed mass also facilitates species dispersal [73,74] and promotes higher longevity in the seed bank [46,47], in such a way that the increase in light and decrease in leaf litter cover promoted by gap formation can enhance the germination of the species from the seed bank [18]. By contrast, plots in which gaps were opened less than ten years ago (gap opening plots) did not significantly differ from plots without gaps in terms of species composition in NMDS1. This might indicate that, while new species arrive at the gap in the first years after gap creation (increasing species richness), the shade-tolerant species originally present before gap opening (i.e., not associated with gaps) still remain in the gap and thus, the understory species composition does not yet change significantly.

In persistent gaps that remained open for at least ten years, the lack of differences in richness with other plot categories is consistent with previous studies, showing that in old-growth forests, the highest species richness is found in recently opened gaps, but as gap age and herbaceous cover increase, forest productivity starts to decrease, leading to resource competition and reducing richness in the understory [12,16,72,75–77]. At this stage, taller plants will have a competitive advantage over shorter plants, replacing them [72].

This is the case of *Urtica dioica*, which has been found to replace *Taraxacum* gr. *officinale* over time [12,72], as seen in our study. In general, the species selected as indicators of gap persistence are related to sites with high light availability (as is the case for *Juncus* gr. *effusus*, *Urtica dioica*, *Veronica officinalis* and *Pilosella* sp.; [71]) and have other dispersal strategies (i.e., achenes without pappus in *Urtica dioica*; [37]) that seem to be less effective than those of early-colonisers or gap opening species. In addition, some of these species are also related to fertile sites with high nitrogen levels (i.e., *Urtica dioica*, *Dryopteris dilatata* (Hoffm.) A. Gray; [71]), which can also be related to gaps, as tree fall leads to an increase in organic matter in the understory and higher light levels promote a high rate of litter decomposition in the soil. Regarding traits, these species tend to have low SLA and LS, which are adaptive responses to high-radiation levels [46,47]. The different values of plant height can be consistent with the higher functional diversity found at sites with intermediate levels of competition [78]. As a result of this process, the understory species composition in NMDS1 in persistent gaps will finally be significantly different from that in plots without gaps. This suggests that changes in understory species composition occur more slowly than changes in the studied environmental conditions, needing more than ten years to really be significant. This is consistent with other studies showing low variation in understory composition between plots with and without gaps [75].

Something similar occurs for plots with gap closure, which despite having a closed canopy at the end of the decade, are still more similar in species composition to plots with gap persistence than to plots without gaps. In this sense, the idea that gaps promote a high variation in species composition that remains even after canopy closure is supported by previous studies [35,36]. Species selected as indicators of gap closure are both related to intermediate levels of nitrogen, but in terms of light, they follow different strategies [71]. *Athyrium filix-femina* is associated with shade [71], as we can also interpret through its leaf traits [46,47], and this is consistent with the lower levels of light found as a result of gap closure. However, the presence of heliophyte species such as *Campanula patula* [71], which remain there even after the gap was closed, might slow down the change in species composition towards understory communities typical of closed canopy, as previously found. Again, more than ten years are needed for a community with a species composition adapted to gaps to return to the original composition characteristic of closed canopy. Despite their influence on understory species composition and richness, at the temporal scale, studied gap dynamics do not seem to influence temporal changes nor final values of species diversity and understory functioning, at least at the community level.

In summary, in the first years after gap creation, new species establish in the gap, increasing species richness, but the shade-tolerant species originally present before gap opening still remain in the gap. Thus, species composition changes slowly over time, needing more than ten years to show significant changes. A similar process happens with gap closure but in the opposite sense. This process of gap dynamics might result in the conservation of the heterogeneity of the understory, and thus, in the maintenance of the whole forest diversity. Indeed, the significantly lower changes in species composition found between plots with persistent gaps and without gaps towards NMDS2 might indicate that persistent gaps are acting as stabilising units, slowing down the process of canopy closure dominant in this forest and the subsequent homogenisation of understory taxonomical composition over time towards species more typical of humid and shaded sites. In this sense, further studies might be necessary to study in more detail how long the vegetation found in persistent gaps might remain there after canopy changes. In addition, not only the temporal evolution of gaps but also other characteristics such as gap size or shape, and even the position of the plot within a gap, appear to have an important influence on forest regeneration and its understory [27,28]. Thus, future studies should also take into account the effect that different gap characteristics might have on understory species and regeneration patterns in order to provide a detailed insight into the forest understory response to gap formation.

## 5. Conclusions

Through their influence on light availability and leaf litter cover, gap dynamics influence the species composition and richness of the understory. The temporal evolution of gaps influences the forest understory, with different species being selected over time following gap evolution. The selection of a certain species depends on its functional traits, among which shade tolerance and dispersal strategies are the most important ones. Changes in understory species composition in response to canopy changes occur at a slow speed in old-growth temperate forests, needing more than a decade to really be significant. Gap dynamics and especially the creation of persistent gaps favour the maintenance of the environmental, taxonomical and functional heterogeneity of the understory of temperate forests, enhancing their stability and slowing down the homogenising effects of canopy closure by beech growth.

**Supplementary Materials:** The following are available online at https://www.mdpi.com/article/10.3390/f12070938/s1, Table S1: Correlation between species abundances in 2006 and 2016, Table S2: List of species and trait values, Figure S1: Pictures of plots with no gaps and with gaps, Figure S2: Aerial photographs of three time periods (1991–1995, 2006, 2017), Table S3: List of environmental variables, Table S4: Datasets from TRY database, Figure S3: Ordination plot with correlated variables, Table S5: Correlation between ordination axes and environmental variables, Table S6: Environmental differences between plots without and with gaps, Figure S4: Understory changes between years and plot categories.

**Author Contributions:** Conceptualization, R.I. and A.G.; Data curation, R.I. and A.G.; Formal analysis, M.V.; Funding acquisition, M.V., R.I. and A.G.; Investigation, M.V., R.I. and A.G.; Methodology, M.V., R.I. and A.G.; Project administration, R.I. and A.G.; Supervision, R.I. and A.G.; Visualization, M.V.; Writing—original draft, M.V.; Writing—review and editing, R.I. and A.G. All authors have read and agreed to the published version of the manuscript.

**Funding:** This research was funded by Fundación Universitaria de Navarra (research Project "Estudio de monitorización integrada en una cuenca forestal del Pirineo"). M.V. is supported by Departamento de Educación, Gobierno de Navarra (Ayudas predoctorales para la realización de programas de doctorado de interés para Navarra; Plan de Formación y de I+D 2018). A.G. is supported by project FORMAL (ref. RTI2018-096884-B-C31; Spanish Ministry of Science, Innovation and Universities).

**Data Availability Statement:** The data presented in this study and the R code used are openly available in the Dryad Digital Repository: https://doi.org/10.5061/dryad.gf1vhhmqb.

**Acknowledgments:** We are grateful to the Parque Natural "Señorío de Bertiz" for allowing us to conduct this research within the protected area. We would also like to thank Javier Puy for measuring radiation and all the authors (see Table S4) that allowed us to use their datasets from the TRY database.

**Conflicts of Interest:** The authors declare no conflict of interest. The funders had no role in the design of the study; in the collection, analyses, or interpretation of data; in the writing of the manuscript, or in the decision to publish the results.

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
