# Peer review of "The Role of Canopy Cover Dynamics over a Decade of Changes in the Understory of an Atlantic Beech-Oak Forest"

_forests, doi:10.3390/f12070938_

Round 1

Reviewer 1 Report

This manuscript by Valerio et al. presents data from two inventories, the first in 2006 and the second in 2016, of 102 forest plots in Bertiz Natural Park, Spain. Changes in understory community composition and their relationship with gap dynamics were analyzed and discussed.

I have a few comments and suggestions for the authors to consider in their revision to improve this study.

First, my biggest confusion is why some shade-tolerant species were identified as indicator species for ‘gap opening’ plots. The current discussion cannot convince me. It is well established that the increase of light availability favors light-demanding species in newly formed gaps. If it is the case that ‘the shade-tolerant species originally present before gap opening still remain in the gap’ as you wrote on line 407, then these species must also be common in ‘no gap’ plots – they cannot be indicators of ‘gap opening’.

Second, the authors are talking about aspects (e.g., south-facing plots on line 285) in results section 3.2, but I can not find in neither the methods nor the tables/figures how this was arrived. Relevant to this issue, I would suggest the author to include a figure about the study area, showing the spatial distributions of the plots, plus a digital elevation model. This way, readers not familiar with Spain and Bertiz forest could have a more intuitive idea of the study design.

Third, I thank the authors for willing to archive the data in an open repository. But I would suggest the authors also to provide the R code used in this study, to further support open and reproducible research.

Minor points:

Table 2: please interpret what are the statistics.

L360: what are in bold?

Reviewer 2 Report

The manuscript “The role of gap dynamics over a decade of changes in the understory of a temperate forest” is a study linking the changes in plant community to changes in canopy cover. The manuscript is well written and the statistical analyses seem properly conducted. However, I have a doubt if the design of the study enables us to link the changes in the plant community to the presence/absence of a gap. The authors designed a plot as 400 m2 quadrats and defined canopy gaps as openings in the canopy layer with a canopy cover lower or equal to 80% citing the paper of Kern et al. 2013. However, Kern et al. created experimental gaps with specific diameters in their study design. Thus, I could not find any relation between their study designs. Moreover, it is broadly accepted to use the definition proposed by Runkle (1982) in studies about canopy gaps. Thus, a canopy gap is defined as an opening in the canopy created by the death of at least one tree from canopy layer. The authors didn’t mention this definition in their methods. The opening in the canopy (or the lower canopy cover) can be caused by some topographic factors (stream, valley), instead of a tree death. It should be specified in the MS what definition was used. Did you try to identify what was the cause of lower canopy coverage (i.e., did you check for the remains of dead tree)? Moreover, 80% canopy coverage of 400 m2 plot gives 80 m2, which is much larger than the lower threshold used to describe a canopy gap in most canopy gaps related studies (usually from 5 to 20 m2), (see: Nagel & Svoboda 2006; Zeibig et al. 2005; Bottero et al. 2011; Orman & Dobrowolska 2017). The authors did not provide any information about the distribution of canopy gap sizes in their research area. I think that a size of gaps may have a significant effect on plant regeneration but this variable was not use anywhere in the study. Since the authors checked plant species composition on regular grid, we actually don’t know if 80 m2 of canopy opening is a small canopy gap or it could be a periphery of large gap. Such variance in canopy gap sizes and a position of their research plot within a gap, should also be consider in their study. In conclusion, I think it is safer to write about the changes in canopy coverage in this study, instead of a presence/absence of a gap.       

Title – A temperate forest is a broad expression. I would suggest you to narrow it down to the forest type in which you conducted your study. Consider to change gap dynamics to canopy cover dynamics.

L33 What do you mean by the most representative? Please specify.

L66 Citation is needed. For recruitment release see: Nagel et al. 2010; Orman et al. 2021.

L68 The size of gap, gap shape, cardinal points inside a gap and a distance from a gap border are factors which also matters. Moreover, these gap characteristics may affect tree regeneration differently depending on life stage. Please see: Nagel et al. 2010; Rozenbergar et al. 2007; Garbarino et al. 2011; Orman et al. 2021.

L78 The speed of lateral growth depends greatly on main tree species in the canopy. Beech is known as a very plastic species which is able to close the canopy in relatively short time (see Schröter et al. 2011). However, that is not the case with conifers.   

L111 Have you conducted any spatial analysis of gap distribution so you can claim a random distribution?

L120 & L137-143 If you identify a plot as a canopy gap when the canopy cover was 80% or lower, you got a huge range of possible gap sizes (80-400 m2). I believe it is meaningful to include a gap size as a variable which may affect understory plant composition. I also wonder if you did not see any difference in spatial distribution of plants on such big area as 400 m2.

L138-139 Did you identify a canopy gap based on % of canopy cover measured by crown projection on a plane? According to the gap definition by Runkle (1982), we consider a gap as an opening in the canopy created by the death of at least one tree. In other words, you may have a less dense canopy and don’t have a gap in it. Please specify if that was the case.

L152 How did you chose the place of measurement? Please specify.

L202 Please provide the detailed formula

L210 What is SES

L220 Why did you decide to add only Quercus saplings but not beech saplings to your analyses?

L476 Why did you mention only beech lateral growth if the forest was composed of oak as well?

Table 1 – Please provide the detailed description of all abbreviations you used so the readers don’t have to scan the MS to understand the table content.

Round 2

Reviewer 2 Report

Many thanks for your careful revision. In my opinion the article has improved a lot and it is ready to be published.